# In Vitro and In Vivo Models of CLL–T Cell Interactions: Implications for Drug Testing

**DOI:** 10.3390/cancers14133087

**Published:** 2022-06-23

**Authors:** Eva Hoferkova, Sona Kadakova, Marek Mraz

**Affiliations:** 1Molecular Medicine, CEITEC Masaryk University, 62500 Brno, Czech Republic; 2Department of Internal Medicine, Hematology and Oncology, University Hospital Brno and Faculty of Medicine, Masaryk University, 62500 Brno, Czech Republic; 3Faculty of Science, Masaryk University, 61137 Brno, Czech Republic

**Keywords:** chronic lymphocytic leukemia, T cells, models, CD40L, IL-4, IL-21, interleukin, venetoclax, ibrutinib, fludarabine, B cells, interactions, microenvironment, therapy resistance, co-culture, xenograft, Eμ-TCL1

## Abstract

**Simple Summary:**

Chronic lymphocytic leukemia (CLL) cells in the peripheral blood and lymphoid microenvironment display substantially different gene expression profiles and proliferative capaci-ty. It has been suggested that CLL–T-cell interactions are key pro-proliferative stimuli in immune niches. We review in vitro and in vivo model systems that mimic CLL-T-cell interactions to trigger CLL proliferation and study therapy resistance. We focus on studies describing the co-culture of leukemic cells with T cells, or supportive cell lines expressing T-cell factors, and simplified models of CLL cells’ stimulation with recombinant factors. In the second part, we summarize mouse models revealing the role of T cells in CLL biology and implications for generating patient-derived xenografts by co-transplanting leukemic cells with T cells.

**Abstract:**

T cells are key components in environments that support chronic lymphocytic leukemia (CLL), activating CLL-cell proliferation and survival. Here, we review in vitro and in vivo model systems that mimic CLL–T-cell interactions, since these are critical for CLL-cell division and resistance to some types of therapy (such as DNA-damaging drugs or BH3-mimetic venetoclax). We discuss approaches for direct CLL-cell co-culture with autologous T cells, models utilizing supportive cell lines engineered to express T-cell factors (such as CD40L) or stimulating CLL cells with combinations of recombinant factors (CD40L, interleukins IL4 or IL21, INFγ) and additional B-cell receptor (BCR) activation with anti-IgM antibody. We also summarize strategies for CLL co-transplantation with autologous T cells into immunodeficient mice (NOD/SCID, NSG, NOG) to generate patient-derived xenografts (PDX) and the role of T cells in transgenic CLL mouse models based on TCL1 overexpression (Eµ-TCL1). We further discuss how these in vitro and in vivo models could be used to test drugs to uncover the effects of targeted therapies (such as inhibitors of BTK, PI3K, SYK, AKT, MEK, CDKs, BCL2, and proteasome) or chemotherapy (fludarabine and bendamustine) on CLL–T-cell interactions and CLL proliferation.

## 1. Introduction

The biology of chronic lymphocytic leukemia (CLL) B cells largely depends on microenvironmental interactions providing pro-survival and pro-proliferative signals [1]. In immune niches, CLL B lymphocytes are surrounded by other immune cells, which create tumor-supporting microenvironments, and these interactions are required for CLL-cell proliferation, which mainly occurs in the lymph nodes [2,3,4,5]. The interactions with T cells seem to be among the most essential CLL proliferation (co)activators [6,7,8] (Figure 1). This is underscored by the observations that proliferating CLL cells in vivo are located close to activated helper T cells [9,10,11] and that in vitro CLL cells virtually never spontaneously divide [12,13] until they are provided with factors produced by helper T cells, such as CD40L, IL21, and IL4 [11,14,15]. Moreover, changes in the T-cell repertoire have been associated with CLL prognosis [14] and are affected by targeted therapy [15,16]. The T cells of patients with CLL also reveal functional differences with those of the healthy population [17], such as the relative sizes of T-cell subsets [18], the presence of specific clonotypes [19], exhausted phenotypes [20], and lower CD40L levels [7]. T cells/T-cell factors can both sustain survival and CLL-cell proliferation [21,22], but also induce their death if the CLL cells are “over-stimulated” or exposed to T-cell factors in the context of the wrong stimuli [23,24]. CLL cells in immune niches produce T-cell-attracting chemokines, such as CCL22 and CCL3/4, and thus directly modulate the immune niches towards T-cell help [25,26]. This further suggests that CLL cells welcome T-cell help, and this is synchronized with BCR activation, which directly triggers CCL3/4 release by CLL cells [26,27]. The specific effects of individual T-cell subsets (beyond CD4^+^ vs. CD8^+^) [8,28] are not generally considered in co-culture CLL–T-cell models, as these models require a relatively large number of T cells, and specific subsets would be difficult to isolate from the blood sample in sufficient quantities. Nevertheless, it has been suggested that follicular helper T cells might be best able to support CLL proliferation [9,29] (Figure 1).

Density-gradient-separated peripheral blood mononuclear cells (PMBCs) from CLL patients are often considered a “pure” population of CLL cells, even though they typically include a small proportion of healthy T cells, B cells, NK cells, monocytes, and dendritic cells. Notably, the CLL-cell viability in such PMBCs cultures tends to be slightly higher when compared to highly purified CLL cells from the same patient cultured in vitro [30]. However, CLL cells do not proliferate in PBMCs cultures, which seems to reflect the situation in the peripheral blood, where activated Ki67^+^ CLL cells usually account for less than 5% of the CLL population. The proliferative CLL cells in the peripheral blood are mostly restricted to the subpopulation that has recently left the immune niches (CXCR4^dim^CD5^bright^ cells) [4,31,32,33].

These observations, along with the lack of CLL-cell proliferation in vitro, allow researchers develop various in vitro and in vivo models to utilize T-cell factors for CLL-cell proliferation and signaling studies. Here, we summarize these models, which have a variable degree of complexity and differ in practical experimental aspects, such as reproducibility, cost, and technical feasibility. Some models also integrate non-T-cell factors, such as stromal cells, nurse-like cells [29], or Toll-like receptor stimulation [34] (Figure 1). In this review, we progress from in vitro towards in vivo models, including the use of transgenic mice and immunodeficient animals transplanted with primary CLL cells.

## 2. In Vitro: Co-Culture with T Cells

Autologous T cells directly obtained from the peripheral blood of CLL patients do not induce strong CLL-cell activation, which would result in proliferation. Their pro-proliferative effect in vitro depends on their pre-activation by anti-CD3 and/or anti-CD28/IL2 [35]. For this purpose, both autologous and allogeneic T cells (from non-related CLL patients or healthy donors) can be utilized; nevertheless, their co-culture requires direct cell–cell contact [35]. The ratio between cultured T and CLL cells is applied empirically and varies from prevailing T cells in short-term culture [33] to CLL cells exceeding the number of T cells by 2–5× (Table 1).

CLL cells respond rapidly to the presence of activated T cells, including NF-κB activation within minutes and activation-marker upregulation within several hours (CD38 [36], CD95, or CD86 [37]). Morphological changes (such as increased CLL-cell size, granularity, and clustering) begin to emerge in a one-day co-culture. This is followed by the induction of CLL proliferation, in most cases within ~3 days of co-culture, as assessed by various methods, such as thymidine incorporation assay [35], CSFE dilution [36,38], or cell-cycle monitoring (Table 1). Co-culture with T cells can be maintained for at least two weeks, but with a progressive degree of apoptosis of activated CLL cells [39], suggesting that there is a balance between the rate of proliferation and CLL-cell apoptosis. Moreover, T cells co-cultured with CLL cells eventually express death-inducing molecules (e.g., FasL) and, thus, potentially directly trigger CLL-cell death.

**Table 1 cancers-14-03087-t001:** In vitro models utilizing co-culture of CLL cells with primary T cells.

Stimulatory T Cells	T:CLL Cells Ratio	Duration [Days]	Purification	CLL Viability	CLL Proliferation	Ref.
irradiated autologous/allogenic CD4^+^ T cells activated by anti-CD3, CLL co-stimulation by IL2 and IL10	-	3 (up to 9)	CLL	~100% (~50% *)	[^3^H]TdR uptake, proliferation dependent on T–CLL cell contact	[35]
autologous activated CD3^+^ T cells	1:3	14	CLL	5–20% (5–20% *)	-	[39]
autologous activated CD3^+^ T cells	1:4	4–6	PBMCs CD3-depleted	not affected	CSFE highly variable; absolute increase in number of CD19^+^ cells	[36]
autologous non-activated T cells	same as in original PBMCs	7	PBMCs	~75% (~50% *)	-	[12]
autologous T cells and CD40L- expressing mouse fibroblasts	-	54–154	PBMCs	-	Ki67 (significantly increased at day 14 compared to day 0)	[40]
autologous activated CD4^+^ T cells	1:5	2–6	CLL	-	Ki67 (day 2; 2% vs. 1% in control), CSFE (day 6; 3% vs. 2% in control)	[9]
autologous activated T cells	2:1	1 or 3 h	CLL	-	-	[33]

* Results from control CLL samples cultured at standard conditions are in brackets.

These co-culture experiments allow many fundamental events to be studied, such as CD40 signaling regulation [33], telomerase activity induction [40,41], and anti-apoptotic molecule induction/repression [42]. Notably, the gene expression profile of CLL cells co-cultured with T cells corresponds to the gene expression changes induced by CD40L [38]. This suggests, together with other data, that CD40 is a key factor in CLL–T-cell contact, similar to normal B–T-cell interactions.

Altogether, CLL–T-cell co-culture is a technically demanding system that requires large numbers of primary T cells and their pre-activation in vitro. Primary T-cell use remains an important tool with which to study autologous CLL-derived T cells and an overall reference for the simplified models described below.

## 3. In Vitro: Mimicking T Cells by Cell Lines and Recombinant Factors

Activated T cells produce a wide spectrum of factors stimulating normal B cells [43]. However, it seems that none of these factors alone is sufficient to induce considerable CLL proliferation in vitro [44]. A primary candidate pro-proliferative T-cell factor is CD40L, based on many observations in normal and malignant B cells [45,46,47,48]. Even though the CD40 pathway is reliably activated by soluble recombinant CD40L [49], the pro-proliferative effect is significantly more potent when the ligand is displayed on the cell membrane, bound to beads, or applied as a trimerized ligand [50,51] (Table 2). Some studies have shown that trimeric anti-CD40L monoclonal antibody promotes CLL proliferation [44,52], but only in a fraction of patient samples. Therefore, it is likely that supportive cell lines expressing membrane-bound CD40L provide a stronger and more physiologically relevant signal than the recombinant soluble CD40L [51]. This is utilized in a co-culture model with CD40L transfected fibroblasts, such as a 3T40L cell line derived from NIH/3T3 mouse cells (Table 2). 3T40L co-culture reduces spontaneous CLL apoptosis [23,48] by upregulating anti-apoptotic proteins [1]. Moreover, simultaneously culturing CLL cells with autologous T cells and 3T40L sustains long-term CLL survival [40]. Other stable cell lines, such as HeLa cells expressing CD40L [23], are used less frequently because their fast growth complicates co-culture when used without mitotic inactivation. CD40L-engineered cell lines’ clear advantage is their ability to regulate the amount of presented CD40L by creating clones with various levels from sorted cells, which is not feasible with primary T cells [51,53]. The effects of CD40L in CLL can be also substantially enhanced by activating JAK/STAT via interleukins [50] or B-cell receptor stimulation [44].

Interleukin 4 (IL4) supports the in vitro survival of B cells by itself, but without inducing their division [32,54], which depends on pre-activating the CD40 pathway or stimulating it in parallel [51,52]. Notably, IL4 induces anti-apoptotic proteins in CLL cells, such as MCL-1 [54]. Furthermore, IL4 promotes the expression of surface IgM [55,56] and CD20 [57], which are both relevant for CLL physiology. Combined with PKC stimulation, IL4 leads to CLL cells’ plasmocytic differentiation to IgM-producing cells [58]. In CLL cells, a combination of CD40L and IL4 usually only induces only weak proliferation [42,59], but this can be increased in CLL with unmutated IGHV by the concomitant stimulation of a Toll-like receptor via CpG [34,60] or adding T-cell-derived cytokines, such as IL2 and/or IL10 [61], Table 3. In contrast to interleukins, the CpG’s pro-proliferative effect does not depend on CD40 activation, since CLL proliferation has also been observed in cultures with CpG+IL2 [62] and CpG+IL15 [63,64].

Interleukin 21 (IL21) shows a pro-apoptotic effect in quiescent CLL cells [38] and B-lymphomas [65], but strongly potentiates the effects of CD40L/IL4 stimulation and induces CLL-cell division in most cases, independent of the clinico-biological characteristics of the patient cells [11,38,66,67]. Significantly higher proliferation can be achieved by adding IL21 after stimulating CLL cells with CD40L/IL4 [44] for one day, probably due to the induction of the IL21 receptor via CD40L [68] (Table 3). The effective dose of IL21 for co-culture is ~20 ng/mL, but higher doses (80 ng/mL [66]) might also be relevant in vivo, since it is difficult to determine the local concentration during cell–cell contact in immune niches. Stimulating BCR with soluble anti-IgM significantly improves CLL proliferation in the CD40L/IL4/IL21 system (with trimeric recombinant CD40L), but this does not reach the proliferation rates induced in normal B cells (~50% CLL-cell proliferation after 6 days vs. ~90% normal B-cell proliferation after 4 days) [44].

CD40L-engineered supportive cell lines can be utilized as initial “platforms” for preparing cell lines expressing a variety of factors simultaneously. This approach has not yet been used in CLL, but was successfully applied in mimicking a germinal center reaction for mouse B cells using BALB/c3T3 fibroblasts expressing CD40L and BAFF [69], or in expanding primary diffuse large B cell lymphoma cells by a co-culture with HK (human kidney) cells engineered to express CD40L and IL21 [70].

Other T cell-produced factors are used for CLL cultures less frequently than CD40L-based systems. For example, CLL-cell treatment with another T-cell-produced factor, namely interferon γ (IFNγ), decreases spontaneous apoptosis [71]. Cultures with the interferons IFNβ or IFNγ also support the survival of CLL cells exposed to ibrutinib [72], and this is probably related to the activation of STAT3 signaling and the induction of MCL-1 levels [73]. Anti-apoptotic effects have also been described for IL13 and IL6, produced by Th_2_ cells [74,75]. IL2 and IL10 support CLL proliferation [61] and provide mild CLL viability improvement [76]. It has been reported that CLL cells produce IL10, which might represent an autocrine signaling loop [77]. In summary, CLL-cell co-culture with cells engineered for CD40L expression combined with interleukins (mainly IL4 and IL21) can be used to induce CLL-cell proliferation and study resistance mediated by T–CLL interactions. The source of CD40 stimulation is important to gain predictable and robust responses, as weaker signals (as with the use of soluble ligand) may result in more diverse proliferation-rate responses.

Co-culture models enable the development of complex in vitro models, which can include T-cell-derived factors, but also other microenvironmental stimuli (Figure 1). For example, it has been shown that stromal cells are sources of relevant integrin signaling and produce a variety of cytokines, such as CXCL12 or IL6. Monocyte-derived nurse-like cells produce anti-apoptotic cytokines, such as BAFF and APRIL, which can also be combined with CD40L-based co-culture systems to support CLL survival (Figure 1). Moreover, the addition of Toll-like receptor stimulation via CpG-ODN seems to induce CLL activation and proliferation to an extent similar to CD40L (Table 3) [64,78].

**Table 2 cancers-14-03087-t002:** In vitro models utilizing co-culture of CLL cells with engineered supportive cell lines and soluble factors.

Stimulatory Cells	Stimulatory Soluble Factors (Concentration)	Duration in Days	Purification	CLL Viability	CLL Proliferation	Ref.
irradiated mouse CDw32 L cells	anti-CD40 mAb (0.5 µg/mL), IL4 (500 U/mL)	10	CLL		[^3^H]TdR uptake 10× higher (vs. control without IL4); increased cell count (until day 8)	[79]
CDw32 L cells	anti-CD40 moAb (0.5 µg/mL), IL2, IL10	3 (up to 9)	CLL	~80% (~50% *) at day 9	weak [^3^H]TdR uptake compared to healthy B cells	[35]
non-irradiated 3T40L cell line	a confluent layer of 3T40L	3	?	89% (71% *)	-	[21]
irradiated 3T40L cell line	1:17 (3T40L:B cells)	3	CLL	77% (32% *)	-	[80]
3T40L cell line	-	1–3	PBMCs/ CLL	upregulation of genes promoting CLL-cell survival and cell-cycle arrest		[48]
irradiated 3T40L cell line	IL4 (20 ng/mL)	6	CLL	35% (55% *)	cell cycle (6% cells in S-phase vs. 3% in control); [^3^H]TdR uptake	[42]
allogenic human bone marrow stromal cells	CD40L (1 μg/mL), IL2 (100 U/mL), IL10 (10 ng/mL)	3.5	CLL	35–75% (10–30% *)	BrdU cell cycle (proliferation induced in 8 of 21 samples); ~4% of cells in S+G2/M phase (DAPI staining)	[61]
irradiated 3T40L cell line	CpG (1.5 μg/mL)	3	PBMCs	75% (IGHV unmut.), 70% (IGHV mutated)	CSFE (proliferation in 80% IGHV unmutated CLL, 25% IGHV mutated CLL at day 5)	[60]
mouse fibroblasts expressing CD31	IL4 (5 ng/mL)	7	PBMCs	60% (30% *)	CSFE; 4–10% Ki67^+^ cells	[59]
mouse fibroblasts expressing CD40L	IL4 (5 ng/mL)	7	PBMCs	50% (30% *)	CSFE; 8–14% Ki67^+^ cells	[59]
3T40L cell line	IL21 (25 ng/mL)	5	CLL	~80% (~30% *) at day 8	CSFE (~50% divided cells, ~20% w/o IL21); Ki67	[38]
non-irradiated human-bone-marrow stromal (BMSC) cell line UE6E7T-2	CD40L (1 μg/mL), CpG (1.5 μg/mL), 1:100 (BMSC:B cells)	3	PBMCs	25% (48% *); 32% (60% *)	↑ S phase; 11% Ki67^+^ cells (1.18%); ↑ S phase, 7.68% Ki67^+^ cells (0.81%)	[34,81]
irradiated mouse L cells expressing CD40L	IL21 (12.5 ng/mL)	9–10	CLL	94% at day 5	CSFE (proliferation of 49% cells; 18% in control with IL4 instead of IL21)	[67]
non-irradiated human BMSC cell line UE6E7T-2	CD40L (1 μg/mL), CpG (1.5 μg/mL), anti-IgM (10 μg/mL)	2	PMBCs	137% (100% *)	7% Ki67^+^ cells	[82]
3T40L cell line	IL21 (25 ng/mL)	5	PBMCs	as control (~90% *)	CSFE (increased division ~10× compared to control)	[78]
HS5 cell line	IL2 (50 ng/mL), CpG (1 µg/mL)	4	PBMCs	70% (82% *)	CSFE (proliferation of 30% cells)	[62]
BMF cell line ^#^	CpG (2 µg/mL), IL15 (10 ng/mL)	7	CLL	-	CSFE (several generations); Ki67	[63]

* Results from control CLL samples cultured at standard conditions are in brackets. ^#^ Generated from a long-term culture of bone marrow cells from a CLL patient. ↑ means an increase in percentage of cells in the given cell cycle phase

**Table 3 cancers-14-03087-t003:** In vitro models utilizing CLL-cell stimulation with recombinant factors.

Recombinant Factor (Concentration)	Duration in Days	Purification	CLL Viability	CLL Proliferation	Ref.
trimeric anti-CD40L moAb, IL4 (20 ng/mL)	4	PBMCs	-	[^3^H]TdR uptake (56% of cells with a high rate of DNA synthesis)	[50]
trimeric anti-CD40L moAb (0.5 μg/mL)	3	PBMCs	55% (22% *)	cell cycle (5.3% of proliferating cells); 31% samples non-responding	[52]
IL4 (2 ng/mL), CD40L plus enhancer (100 ng/mL)	7	CLL	~80% (40–80% *)	-	[12]
CD40L or anti-CD40 moAb, IL4 (10 ng/mL), IL21 (20 ng/mL)	1.6	PBMCs	-	[^3^H]TdR uptake; BrdU cell cycle, 25% cells in S+G2/M phase (0.7–7.8% *)	[11]
CpG (5 μg/mL), IL21 (50 ng/mL)	2.3	CLL	no difference from control	[^3^H]TdR uptake, significantly increased proliferation with addition of IL21	[83]
histidine-tagged CD40L (50 ng/mL), anti-polyhistidin mAb (5 μg/mL), CpG (10 μg/mL), IL2 (50 ng/mL), IL10 (50 ng/mL), IL15 (10 ng/mL), and IL6 (50 ng/mL), in specific time-dependent sequence	7	CLL	differentiation into antibody-producing cells		[84]
IL15 (15 ng/mL), CpG (1.5 μg/mL)	6	CLL	60–80%	CSFE (~70% divided cells)	[64]
IL15 (15 ng/mL), CpG (1.5 μg/mL)	5	PBMCs	<5% difference from control	CSFE (significantly increased cell division)	[78]
anti-IgM (10 µg/mL), trimeric CD40L (100 ng/mL), IL4 (10 ng/mL), and IL21 (25 ng/mL)	6	CLL	-	CSFE (41% of CLL samples proliferating)	[44]

* In the brackets are results from control CLL samples cultured in standard conditions.

## 4. In Vitro: Use of CD40L-Based Systems for Drug Testing

Culture systems with T cells or T-cell signals make it possible to obtain a considerable fraction of activated CLL cells with features of lymph-node CLL cells [3,85]. This can be used to test the resistance (or sensitivity) induced by CLL–T-cell interactions (Figure 2).

It has been known for over two decades that stromal cells can provide resistance to classical chemotherapy and some antibody-mediated effects [97,98]. Similarly, CD40 stimulation was shown to provide resistance to fludarabine [80,81] and BCL2/BCL-XL inhibitors (venetoclax [86,99], ABT-737 [89,100]) (Figure 2). Stronger resistance to BH3-mimetics can be achieved by adding IL21 to the culture [78,90]. For such studies, CD40 pathway activation has been achieved by co-culture with activated T cells [101], a 3T40L cell line [99], or soluble CD40 ligand [102].

These studies also revealed that cerdulatinib, a dual SYK/JAK inhibitor, can prevent BCL-XL upregulation via CD40L in the lymph nodes, and this is synergistic with venetoclax [94], ibrutinib [103,104], or the MEK1/2 inhibitor, binimetinib [95]. It has been also shown that ibrutinib indirectly blocks CD40 signaling in CLL cells by reducing TRAF4 levels [33]. This might also explain why ibrutinib is relatively efficient at blocking CLL-cell proliferation since it impairs both BCR signaling and the CD40 pathway (together with other secondary effects) [63].

Notably, CD40 ligation sensitized cells to anti-CD20 antibody rituximab in vitro [92,105] (Figure 2). Furthermore, it has also been shown that CD40 moAb selicrelumab or 3T40L co-culture sensitizes CLL cells to anti-CD20 antibody obinutuzumab [105]. This is contrary to using fibroblast-like stromal cells, such as HS5 (not expressing CD40L), which protect malignant B cells from rituximab’s effects via a mechanism of cell-adhesion-mediated drug resistance [106,107].

The models that reveal the role of CD40 stimulation in drug response frequently contain other concomitant signals, such as interleukins, which modulate the observed effects. IL4 has been shown to induce JAK3-dependent resistance against chemotherapeutics [108] and to mitigate the impact of ibrutinib and idelalisib on BCR signaling [55,109]. IFNγ was found to increase CLL survival via MCL-1 upregulation, which might support drug resistance [73]. On the other hand, the pro-apoptotic effect of IL21 is synergistic with the treatment of malignant B cells with fludarabine or rituximab [110].

An interesting example of a molecule directly affecting T–CLL-cell interactions is the immunomodulatory drug lenalidomide. Lenalidomide has been shown to improve CLL–T-cell synapse formation [111]. Lenalidomide also stimulates both IL21 production by T cells and IL21 receptor overexpression on CLL cells, leading to IL21-induced cytotoxicity [112].

Altogether, these data show that models mimicking CD40 signaling can be especially useful to study drug resistance in CLL, and they suggest that directly targeting CD40 might be of therapeutic interest. Indeed, the anti-CD40 antibody has been investigated in a I/II phase clinical trial on lymphoma (study NCT00670592), with a clinical effect in approximately one-third of follicular lymphoma patients [113].

## 5. In Vitro: CLL Co-Cultures in 3D

Culture conditions based on culture-treated polystyrene plates provide a common planar platform for in vitro studies of CLL. Nevertheless, in vivo interactions are not limited to 2D surfaces and occur in three-dimensional (3D) space. The 3D microenvironment can be mimicked by the use of, for instance, spheroids, scaffolds, or dynamic bioreactors with continuous flow [114]. Three-dimensional and scaffold-based approaches are not yet widely applied in CLL research but can be inspired by models used for hematopoietic stem-cell expansion or bone-marrow-niche modeling. Solid biomaterials can mimic bone-marrow structures, which feature original trabecular architectures and are loaded with supportive cell types, such as mesenchymal stromal cells [115,116,117,118]. The artificial scaffolds can also contain polymers (e.g., PEG or hydrogel-based) or components of bone, such as hydroxyapatite or β-tricalcium-phosphate. These biomaterials are often modified by adding extracellular matrix proteins or peptides (laminin, collagen-derived RGD peptide, and fibronectin) to support the growth of cells.

The main advantage of using biomaterials is the design of a specific topography and structure stiffness [119]. Recently, primary CLL cells were cultured in 3D-printed, commercially available CELLINK Bioink, which maintained 40% CLL-cell viability after one month [120]. It is tempting to speculate that adding CD40L and/or interleukins could improve these 3D models and also induce significant proliferation [121]. It has been shown that T-cell signals can be introduced to scaffolds via injection, the covalent linking of recombinant factors [122], or seeding with supportive cells producing the desired factors. The second option was extensively developed to stimulate healthy murine B cells [123] using supportive cells expressing CD40L and BAFF. These cells functionally substituted follicular T helper and dendritic cells and led to murine B-cell differentiation into GC-like B cells [123]. Moreover, the mitomycin-C-treated supportive cells encapsulated in gelatin with murine B cells form organoids that can be maintained for up to 10 days. Gelatin (cross-linked with silicate nanoparticles) provides integrin signaling, which seems to be important for GC-like reactions. The addition of IL4 to the described co-culture also induces class-switch recombination in 6–8 days, and the organoid structure leads to a murine B-cell expansion of up to 100-fold, compared to a 10–20-fold increase in planar culture [123].

CLL-cell culturing in bioreactors is in its infancy, but this dynamic form of culturing might be of high relevance for CLL-cell recirculation and migration studies between lymphoid organs and the periphery. Walsby et al. [124] used a semi-closed circulatory system propelled by a peristaltic pump with designed hollow-fiber areas coated with endothelial cells, through which CLL cells could migrate. This revealed that migrating CLL cells tend to have lower CXCR4 levels and higher CD5 levels [125]. However, it remains to be determined whether such changes in cell-surface molecules are causes or consequences of migration through the endothelial cell layer deposited on the fibers, since it has been shown that CXCR4^dim^CD5^bright^ CLL cells isolated directly from patients’ peripheral blood have a lower migration capacity when injected into immunodeficient mice [32]. A dynamic system might also mimic CLL interactions in bone marrow by introducing scaffolds coated with bone-marrow stromal cells (HS5 cell line) [126]. Cultures incorporating T cells or even complex lymphoid structures have not yet been reported for CLL, but 3D culture in a microfluidic system enables the in vitro survival of follicular lymphoma primary cells co-encapsulated with tonsil stromal cells [127].

Three-dimensional cultures represent a novel approach in the CLL field, with the potential to introduce various microenvironmental stimuli, including T cells and T-cell-derived factors.

## 6. In Vivo: Role of T Cells in Transgenic CLL Models

Several transgenic CLL models have been developed, with the Eµ-TCL1 murine model being the most widely used [128]. This model is based on the overexpression of the anti-apoptotic *TCL1* gene under the *V_H_*-promoter-*IgH*-Eμ-enhancer, and although this is not directly related to any aberration in CLL patients, it proved to be useful for studies of multiple aspects of CLL biology [128,129]. Eµ-TCL1 mice develop monoclonal or oligoclonal CD5^+^ B cells, and at 13–18 months of age, they manifest splenomegaly, hepatomegaly, and lymphadenopathy [130]. Eµ-TCL1 mice form normal immune systems, including T cells and NK cells; however, the animals gradually develop T-cell defects. Similarly, T-cell defects were noted in the adoptive transfer of Eµ-TCL1 CLL cells from an older animal into young littermates [131] and into wild-type mice [132] (Figure 3).

It remains unclear if CLL development in Eu-TCL1 mice is potentiated by T-cell help. Grioni et al. showed negligible Eµ-TCL1 leukemic clone proliferation in TCL1+/+AB0 mice lacking CD4^+^ T cells. Interestingly, proliferation was not influenced by a lack of CD40L stimulation, as leukemic cells proliferated in wild-type mice treated with anti-CD40L antibody, as well as in mice with no CD40L expression [134]. On the other hand, Kocher et al. found that transplanting Eµ-TCL1 splenocytes led to the shorter survival of GK5 mice, which had a complete loss of CD4^+^ cells, compared with wild-type mice [133] (Figure 3).

An increased propensity to signal CD40 can affect B-cell transformation. This is promoted by the elevated expression of some TRAF-family proteins, particularly TRAF1, which was also found in CLL [136]. TRAF1 forms a heterodimer with TRAF2 and induces the activation of the classical NF-κB signaling pathway downstream of CD40 [137]. Transgenic mice expressing, in lymphocytes, a TRAF2 mutant lacking the RING and zinc finger domains located at the N terminus of the molecule (TRAF2DN) develop the polyclonal expansion of B lymphocytes [138]. Interestingly, TRAF2DN is structurally similar to TRAF1, which is the only TRAF-family member that lacks a RING finger domain. TRAF1 and TRAF2DN can heterodimerize with TRAF2, modulating TRAF2 activities. Zapata et al. [135] showed that transgenic mice expressing both TRAF2DN and BCL-2 in the B-cell lineage develop age-dependent B-cell leukemia and lymphoma, with similarities to human CLL (Figure 3). This underscores the role of TRAF family members in the aggressiveness and CD40 signaling in CLL [33].

## 7. In Vivo: Co-Transplantation of CLL and T Cells in Xenograft Models

Engrafting primary CLL cells into mice and the generation of patient-derived xenograft (PDX) models are long-standing problems in CLL research (Figure 4). The first successful attempts were in studies on lethally irradiated normal mouse strains (BALB/c), which were radioprotected with bone marrow from SCID mice. This limits the growth of Eppstein–Barr virus (EBV) positivity adjacent to healthy B cells after the intraperitoneal (i.p.) application of PBMCs in CLL patients [139,140]. In these studies, mice receiving PBMCs from low-stage CLL patients engrafted T cells preferentially, and these were found in the spleen, but only rarely in the peripheral blood or bone marrow. By contrast, mice receiving lymphocytes from high-stage CLL patients had only very low T-cell engraftment in the peritoneum and spleen, despite receiving injections of the same total number of T cells [139]. Human B and T cells formed follicles in the spleen and lymph nodes in mice injected with high-stage CLL. However, these structures gradually disappeared from the fourth week post-transplantation. Based on these results, the authors speculated that low-stage CLL patients’ T cells proliferate normally in the mouse model, which allows them to reject CLL engraftment [139]. In a follow-up study, Shimoni et al. showed that depleting T cells using anti-CD3 antibody on the day of transplantation led to substantially better CLL-cell recovery from the peritoneal cavity [140], suggesting that T cells limited the growth of CLL cells (Figure 4).

Subsequently, Dürig et al. investigated the engraftment potential of low-stage vs. high-stage CLL patient samples [141] (Figure 4). They intravenously (i.v.) and intraperitoneally (i.p.) injected 100 × 10^6^ PBMCs into irradiated NOD/SCID mice and detected CLL cells after 4 weeks in the peritoneal cavity, spleen, and peripheral blood. According to the authors, the observed lower engraftment of the low-stage disease might have been a consequence of the lower cytokine production by the T cells from those patients. Interestingly, the CLL cells survived better in the peritoneal cavity than in any of the other compartments analyzed. Notably, the CLL cells did not migrate into the bone marrow, despite human CD3^+^ engraftment in the marrow. On the other hand, the murine spleens contained focuses of CD20^+^ cells with or without CD3^+^ cells. The authors suggested that direct T–CLL-cell contact might not be required for CLL cells to survive and proliferate in such a system [141].

T cells’ positive role in CLL-cell engraftment in NOD/SCID mice was first described in Aydin et al.’s study [142] (Figure 4). They found a noticeable correlation between the proliferation of leukemic cells and the expression of CD38, a marker of activated CLL cells. The problem with this model was the insufficient engraftment efficiency, which was probably due to an insufficiently immunodeficient host mouse strain [142].

The use the NSG mouse strain, which lacks its own B cells, T cells, and NK cells, enabled a stable xenograft model based on CLL–T-cell interactions to be established and is now often used in CLL research [143] (Figure 4). The model utilizes very young (four- to-eight-week-old) NSG mice, which are γ-irradiated and injected intratibially or i.v. with CFSE-labeled CLL-PBMCs. Robust CLL proliferation was observed only in animals with an adequately high number of circulating T cells. In most cases, proliferation was blocked by applying anti-CD3 or anti-CD4 antibodies, with the exception of a few cases, in which the PBMCs came from patients with a more aggressive disease. In contrast to previous attempts, the mice were sacrificed not after approximately 2 weeks, but after up to 12 weeks. The cause of death in the transplanted mice was T-cell proliferation and consequent graft-versus-host disease development [143]. In line with this, a lowered number of injected T cells or depleted CD8^+^ cells led to a longer survival for the transplanted mice [145]. The model also enables the study of the spontaneous and treatment-induced clonal dynamics of the disease [147] and the effect of therapy [148]. On the other hand, CLL cells stimulated with T cells might undergo a process resembling plasma-cell differentiation, which probably does not occur typically in CLL patients [149] (Figure 4).

The original NSG model has been improved by using T cells in vitro preactivated by CD3/CD28/IL2 stimulation, which increases engraftment efficiency [144]. The use of preactivated T cells is not further improved by busulfan pretreatment. In this model, the authors describe a higher percentage of dividing CLL cells in the mouse spleens than CLL in the peritoneal cavity, despite the fact that the T cells divided in a similar manner in both sites. The peritoneal cavity of the mice might mimic the situation of quiescent peripheral blood CLL cells in patients [144] (Figure 4).

The CLL mouse model based on T-cell co-transplantation was further introduced into NOG mice [145], which manifested alterations in lymphocytes and the innate immune system and displayed even better engraftment susceptibility to tumor cells than the NSG strain [150]. Notably, these mice exhibited nearly equal engrafted CLL-cell numbers in the spleen after 28 days of transplantation, regardless of the transplanted T-cell numbers, which differed at the time of injection by up to 100-fold between the animals. Complete T-cell deletion blocked the engraftment [146,151] (Figure 4). Notably, a similar dependence on T-cell signals was also found for indolent lymphoma engraftment [152].

Regarding the importance of T cells for CLL-cell survival, xenograft models substantially contributed to the current view. Nevertheless, on one hand, T cells induce CLL proliferation in immunodeficient animals, while, on the other hand, they can cause the death of the animals.

## 8. Conclusions

T cells are emerging as key components of lymph node niches, providing signals for CLL-cell survival and stimulating their proliferation. CLL co-culture with T cells in vitro has been shown to induce gene-expression changes and phenotype shifts in CLL cells residing in the lymph nodes. The principal driver of CLL-cell activation seems to be the CD40 ligands expressed on the T-cell membrane. This is indicated by multiple successful approaches based on CLL-cell co-culture with CD40L-expressing fibroblasts or soluble ligands. This stimulation induces a transcriptional profile comparable to CLL cells directly interacting with T cells. The addition of T-cell-derived soluble factors further modifies the CLL response in co-culture systems, potentiating the effects of CD40 signaling (such as IL4) or driving activated CLL cells to massively proliferate (such as IL21). Nevertheless, reliable co-culture systems for high-throughput assays involving CLL proliferation are still lacking. Interestingly, even the most complex co-culture systems enable only time-limited CLL-cell survival, since CLL cells do not form stable cultures, as is possible for normal B cells stimulated by T-cell factors. This might point to some uncovered biological and functional CLL–T-cell interaction specificities and CLL pathophysiology. Moreover, co-culture systems likely provide CLL cells with supra-physiological signals, and the constant signal presence does not allow CLL cells to modulate their intensity via migration outside of the proliferative centers, as has been postulated in vivo. This is partially solved in 3D culture systems and bioreactors, which offer variable niches and dynamic interactions. Mouse models, in this context, are the most complex systems. They allow the changes in T-cell composition and potential changes in the interactions during the progression of the disease to be characterized.

Models of CLL–T-cell interactions have substantial potential for application in the testing of novel drugs. In contrast to models of stromal-CLL cells that only support CLL-cell survival and drug resistance, the introduction of T-cell factors can induce proliferation and variably increase or reduce resistance to specific drugs. It is desirable for novel inhibitors and drug combinations to be pre-clinically tested in the context of CLL–T-cell interactions.

## Figures and Tables

**Figure 1 cancers-14-03087-f001:**
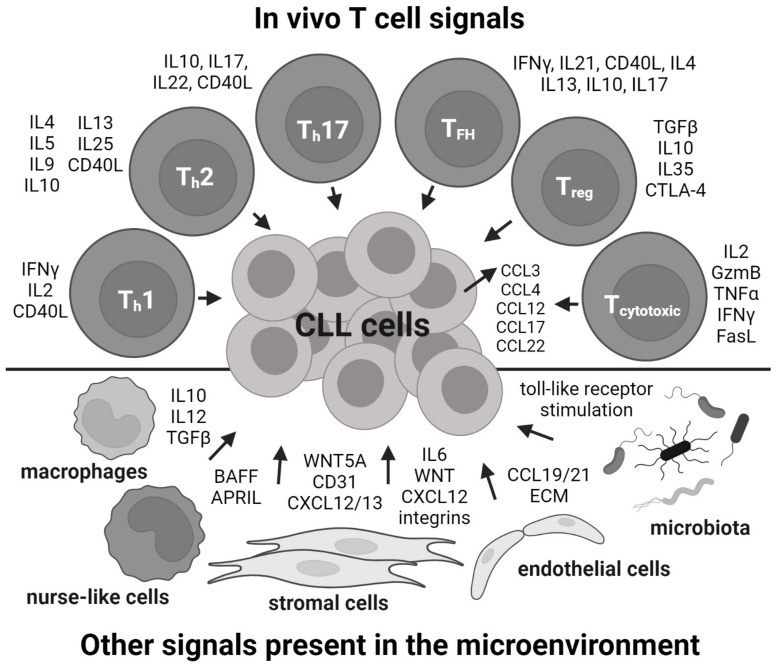
Microenvironmental signals provided by various T-cell subsets and other cell types in immune niches.

**Figure 2 cancers-14-03087-f002:**
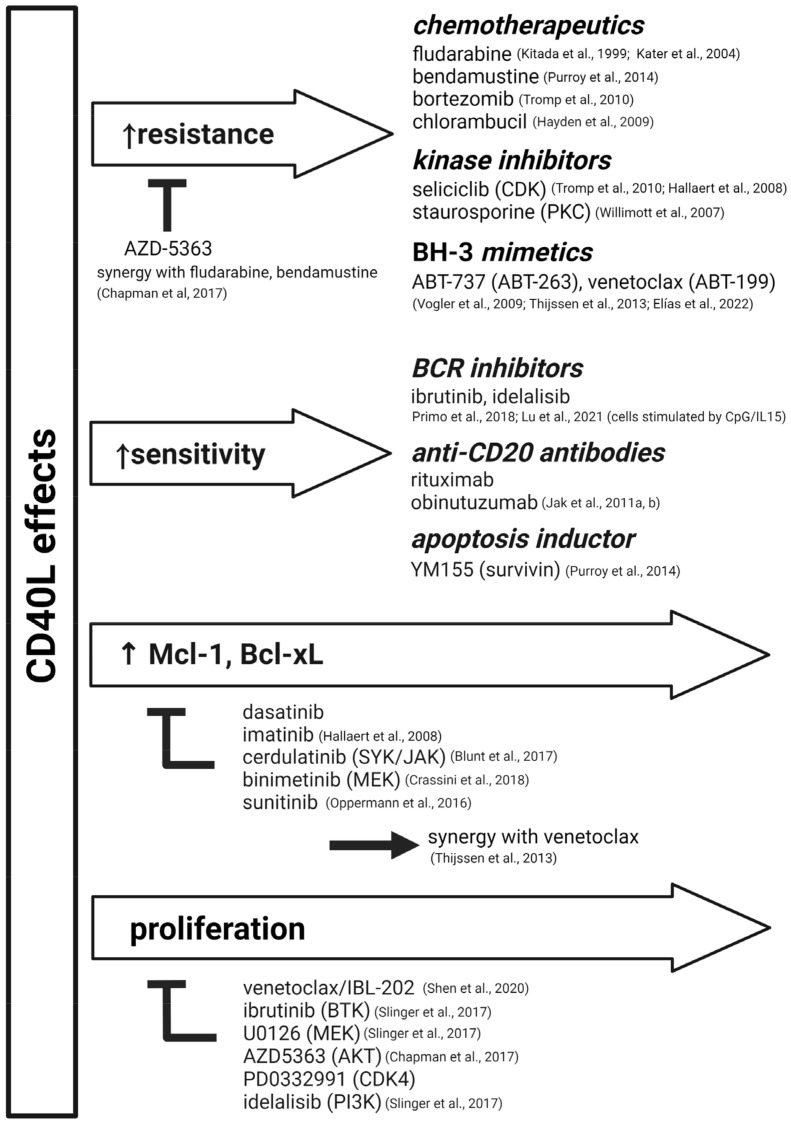
Effects of CD40 pathway on drug responses in CLL cells [21,23,34,42,60,62,63,67,78,85,86,87,88,89,90,91,92,93,94,95,96].

**Figure 3 cancers-14-03087-f003:**
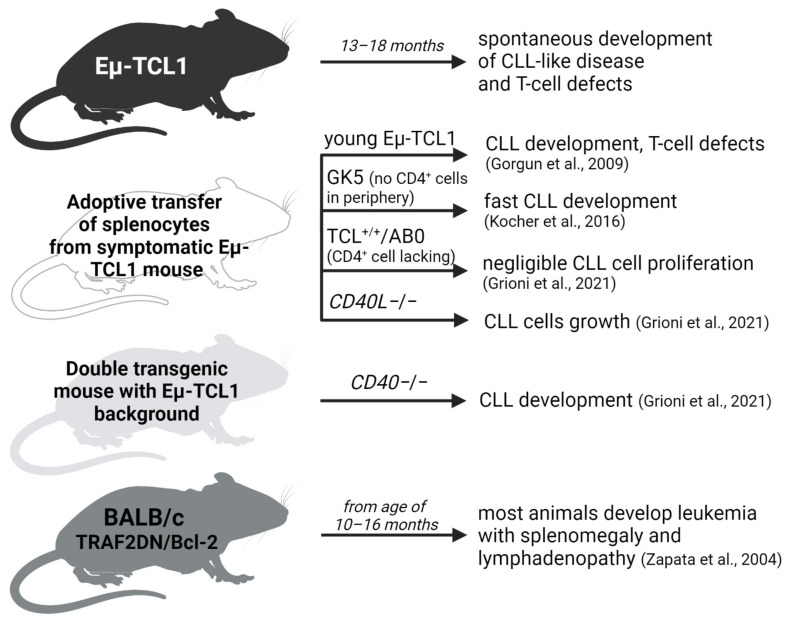
Transgenic mouse models exploring the role of CLL–T-cell interactions [131,133,134,135].

**Figure 4 cancers-14-03087-f004:**
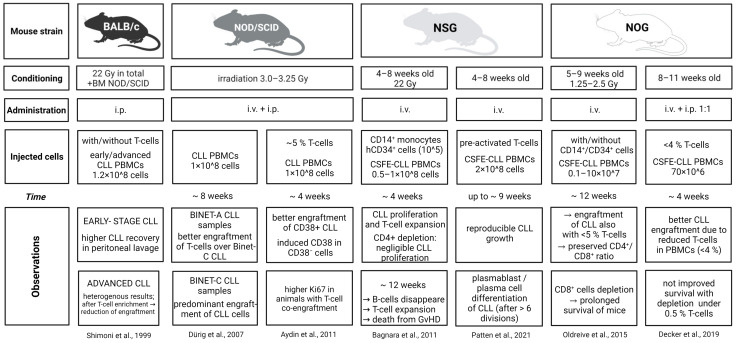
Murine models revealing role of T cells in CLL biology. BM = bone marrow, CSFE = carboxyfluorescein succinimidyl ester, PBMC = peripheral blood mononuclear cells [140,141,142,143,144,145,146].

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
