# Peer review of "In Vitro and In Vivo Models of CLL–T Cell Interactions: Implications for Drug Testing"

_cancers, 2022, doi:10.3390/cancers14133087_

Round 1
Reviewer 1 Report
The review by Hoferkova and colleagues offers a comprehensive overview of the currently available strategies to model CLL-T cell interactions. It is quite dense of information and provides an accurate review of the literature in the field, although mainly focusing on the role exerted by the CD40-CD40L axis. My only suggestion is to improve the figures, maybe adding a cartoon or a sketch recapitulating the variety of signaling molecules relevant in CLL-T cell interactions.
Author Response
Dear Reviewer,
thank you for your time to comment our manuscript and we appreciate your valuable feedback. Here we would like to adress your remark.
Point 1: My only suggestion is to improve the figures, maybe adding a cartoon or a sketch recapitulating the variety of signaling molecules relevant in CLL-T cell interactions.
Response 1: We thank the reviewer for his suggestion. In response, we have added a new figure (now Figure 1) summarising CLL-T cell interactions along with T cell diversity and other signals in CLL microenvironment, which are relevant for in vitro CLL models.
Reviewer 2 Report
In this review Hoferkova and colleagues review models of CLL-T cell interactions. The manuscript is well written and comprehensively reviews the field. A have a few comments that may improve the manuscript
- The review generally discusses bulk T cell populations - some discussion regarding the role of subpopulations of T cells - e.g. bulk CD4+ or T follicular helper, T follicular regulatory cells that may be playing a role could be discussed
- The implications for drug testing focuses on CD40 ligation - are any other T cell derived stimuli implicated in drug resistance?
- While the review clearly focuses on CLL-T cell interactions it is known that other cell types are important in the CLL microenvironment - eg stromal cells, TAMs/Nurse-like cells - this should be discussed as some of proposed models do not account for these
- The in vitro models generally use PB CLL cells which as the authors state are essentially quiescent and not representative of LN CLL cells. Could work be done with the CD5hiCXCR4lo fraction or CLL cells derived from LNs?
- Typo in line 82
Author Response
Dear Reviewer,
thank you for your time to comment our manuscript and we appreciate your valuable feedback. Here we would like to adress your remarks.
Point 1: The review generally discusses bulk T cell populations - some discussion regarding the role of subpopulations of T cells - e.g. bulk CD4+ or T follicular helper, T follicular regulatory cells that may be playing a role could be discussed.
Response 1: We thank the reviewer for his comment. The in vivo role of the individual T cell subsets in CLL has been sumarized in detail in recent reviews (Vlachonikola et al., Front. Immunol. 2021; Roessner & Seiffert, Leukemia 2020; Man & Henley, Br. J. Haematol. 2019). In T cell co-culture models the role of T cell subsets has not been studied and therefore we mainly discuss T cells in general. It is difficult to obtain enough of specific rarer T cell subsets needed for testing in models discussed in the review, which largely prevents such studies. There have been some studies into the effect of other microenvironmental signals on T cell subsets (Vaca et al., Leukemia 2022; Wu et al., Front. Oncol. 2021), but in our review we focus on the technical aspects of the models. To at least partially adress the diversity in T cells, we have added a summarising Figure 1 and a comment in the introduction (lines 53–58).
Point 2: The implications for drug testing focuses on CD40 ligation - are any other T cell derived stimuli implicated in drug resistance?
Response 2: In response to the reviewer's comment, we have added a paragraph into section 4 (lines 221–227).
Point 3: While the review clearly focuses on CLL-T cell interactions it is known that other cell types are important in the CLL microenvironment - eg stromal cells, TAMs/Nurse-like cells - this should be discussed as some of proposed models do not account for these
Response 3: In response to the reviewer's comment, we added a paragraph into the section 3 (lines 178–185), a new figure that summarizes CLL interactions with other cell types (now Figure 1), and also briefly commented this in the introduction.
Point 4: The in vitro models generally use PB CLL cells which as the authors state are essentially quiescent and not representative of LN CLL cells. Could work be done with the CD5hiCXCR4lo fraction or CLL cells derived from LNs?
Response 4: We thank the reviewer for an interesting point. In principle, it is possible to co-culture any sorted CLL subpopulation(s). However, as the differences in CD5hiCXCR4lo cells are induced just by the microenvironmental stimuli (such as IL4 reducing CXCR4, Aguilar-Hernandez et al., Blood 2016) the co-culture will likely diminish the original differences and lead to an activated CLL phenotype after the co-culture. We are unaware of any experimental study that would directly address this question.
Point 5: Typo in line 82.
Response 5: We thank the reviewer for bringing this to our attention.